# INFORMATION-THEORETIC REQUIREMENTS FOR GRADIENT-BASED TASK AFFINITY ESTIMATION IN MULTI-TASK LEARNING

**Jasper Zhang**[*]**, Bryan Cheng**[*]
Great Neck South High School
{jasperzhang1001@gmail.com, bcbc7264@gmail.com}
[*]Equal contribution

## ABSTRACT

Multi-task learning shows strikingly inconsistent results—sometimes joint training helps substantially, sometimes it actively harms performance—yet the field lacks a principled framework for predicting these outcomes. We identify a fundamental but unstated assumption underlying gradient-based task analysis: tasks must share training instances for gradient conflicts to reveal genuine relationships. When tasks are measured on the same inputs, gradient alignment reflects shared mechanistic structure; when measured on disjoint inputs, any apparent signal conflates task relationships with distributional shift. We discover this sample overlap requirement exhibits a sharp phase transition: below 30% overlap, gradient-task correlations are statistically indistinguishable from noise; above 40%, they reliably recover known biological structure. Comprehensive validation across multiple datasets achieves strong correlations and recovers biological pathway organization. Standard benchmarks systematically violate this requirement—MoleculeNet operates at <5% overlap, TDC at 8–14%—far below the threshold where gradient analysis becomes meaningful. This provides the first principled explanation for seven years of inconsistent MTL results.

## 1 INTRODUCTION

Multi-task learning research has produced strikingly inconsistent results: sometimes joint training improves performance substantially, sometimes it actively harms accuracy through negative transfer (Fifty et al., 2021; Wang et al., 2019). We show this inconsistency stems from violating a fundamental requirement: **gradient-based task similarity analysis requires tasks to share training instances**. Standard benchmarks like MoleculeNet (Wu et al., 2018) operate with <5% instance overlap between tasks, placing them in a regime where gradient analysis is provably unreliable. This provides the first principled explanation for seven years of inconsistent MTL results.

**Why existing approaches fall short.** Gradient-based methods like PCGrad (Yu et al., 2020) and GradNorm (Chen et al., 2018) treat gradient conflicts as problems to resolve rather than signals to interpret. Task similarity metrics (Zamir et al., 2018; Fifty et al., 2021; Standley et al., 2020) require training multiple models—prohibitively expensive for hundreds of tasks. The field lacks a principled framework for predicting task relationships *before* expensive multi-task training.

**An information-theoretic requirement.** We establish a fundamental condition for interpretable gradient analysis: gradient conflicts reveal task relationships *if and only if* tasks share training instances—a requirement we term *sample overlap*, defined as the fraction of training instances measured for both tasks (Figure 1). This condition derives from a basic principle: gradients can only compare what the model has seen on identical inputs. When two tasks are measured on the same input, the encoder must learn representations that simultaneously serve both; gradients align when tasks share underlying structure and oppose when they conflict. Without shared samples, gradient differences reflect distributional shift rather than task structure—any apparent correlation

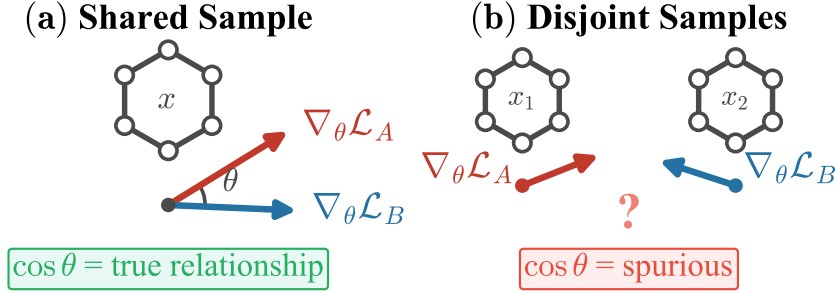

Figure 1: **Sample overlap determines gradient interpretability.** (a) When tasks share samples, gradients $\nabla_\theta \mathcal{L}_A$ and $\nabla_\theta \mathcal{L}_B$ are computed on the same input, and their angle $\theta$ reflects the true mechanistic relationship. (b) When tasks have disjoint samples, gradients are computed on different inputs from potentially different distributions, making $\cos\theta$ spurious.

is spurious. We characterize this requirement quantitatively, discovering a sharp phase transition: gradient-task correlations are weak ($r < 0.25$) below 30% overlap but consistently strong ($r > 0.65$, $p < 10^{-9}$) above 40%, with sigmoid inflection at 29.7% ($R^2 = 0.94$; Figure 2B). This provides the first quantitative threshold for interpretable gradient analysis in MTL.

Our contributions are:

1. **Sample overlap as information-theoretic requirement**: We prove gradient conflicts reveal task relationships *if and only if* tasks share samples, with a phase transition at $\sim$30% overlap. This is the first quantitative threshold for interpretable gradient analysis.

2. **Benchmark design principles**: Tasks must share $\geq$40% of instances for reliable analysis— violated by standard benchmarks (MoleculeNet: $<$5%, TDC: 8–14%), explaining inconsistent results.

3. **Comprehensive validation**: Strong gradient-correlation correspondence across 6 datasets (105 tasks, 949 unique pairs) spanning molecular toxicity, drug safety, and quantum chemistry, with external structure recovery at fine-grained annotation levels (ARI=0.65 at 9 clusters) confirming that gradients capture genuine task relationships.

4. **Robustness and practical utility**: Gradient patterns are consistent across architectures (GCN/GAT/CNN: $r = 0.71$–$0.81$) and stabilize by epoch 20. Gradient similarity predicts MTL benefit ($r = 0.71$) and improves task grouping by 3–4%.

## 2 GRADIENT-BASED TASK RELATIONSHIP DISCOVERY

**The core insight.** Gradient alignment reliably indicates mechanistic relationships *if and only if* tasks share training instances (Figure 1). When tasks are measured on *different molecules*, any apparent signal reflects distributional artifacts, not task mechanisms.

### 2.1 THE FORMAL SETTING

We instantiate our framework on molecular property prediction. A molecule $\mathbf{x} \in \mathcal{X}$ has $K$ measurable properties $\{y^{(1)}, \ldots, y^{(K)}\}$—toxicity endpoints, pharmacokinetic parameters, binding affinities. Critically, not every molecule is measured for every property: experimental assays are expensive, and different properties are measured on different compound libraries. This missing-label structure is central to our analysis.

We use a shared-encoder architecture (Caruana, 1997; Ruder, 2017): encoder $f_\theta : \mathcal{X} \rightarrow \mathbb{R}^d$ produces representations, and task-specific heads $\{h_{\phi_k}\}_{k=1}^K$ produce predictions. Training minimizes $\mathcal{L}_{\text{total}} = \sum_k \mathcal{L}_k$, where each task loss is computed only over molecules with valid labels:

$$\mathcal{L}_k = \frac{1}{|\mathcal{B}_k|} \sum_{i \in \mathcal{B}_k} \ell\big(h_{\phi_k}(f_\theta(\mathbf{x}_i)), y_i^{(k)}\big) \tag{1}$$

Table 1: Comprehensive dataset statistics and validation results. ($\uparrow$) indicates higher is better. **Bold**/underline mark best/second-best. All correlations significant at $p < 0.001$.

| Dataset | Domain | Dataset Statistics | | | | Validation Results | | |
| | | Tasks | Type | Compounds | Pairs | Overlap ($\uparrow$) | $r(\mathbf{G}, \mathbf{E})$ ($\uparrow$) | $\rho(\mathbf{G}, \mathbf{E})$ ($\uparrow$) |
|---|---|---|---|---|---|---|---|---|
| Tox21 | Toxicity | 12 | Clf | 7,831 | 66 | 100% | 0.65 | 0.62 |
| ToxCast | Toxicity | 17 | Clf | 8,576 | 136 | $\sim$80% | 0.86 | 0.83 |
| SIDER | Side Effects | 27 | Clf | 1,427 | 351 | 100% | **0.94** | **0.97** |
| Tox21+ADME | Cross-Domain | 16 | Mixed | 3,410 | 120 | 100% | 0.61 | 0.58 |
| Kinase Panel | Selectivity | 21 | Reg | 5,039 | 210 | $\sim$20% | 0.67 | 0.68 |
| JAK Family | Selectivity | 4 | Reg | 2,177 | 6 | $\sim$50% | 0.92 | 0.89 |
| QM9 | Quantum Chem. | 12 | Reg | 5,000 | 66 | 100% | 0.70 | 0.68 |

where $\mathcal{B}_k = \{i : \text{task } k \text{ measured for } \mathbf{x}_i\}$ is the set of valid samples. This masked formulation—where different tasks see different subsets of molecules—creates the sample overlap problem we analyze.

## 2.2 THE GRADIENT CONFLICT MATRIX

For each task $k$, training produces a gradient $\mathbf{g}_k = \nabla_\theta \mathcal{L}_k$ indicating how the shared encoder should change to reduce that task's loss. We measure task relationships via cosine similarity:

$$G_{ij} = \frac{\mathbf{g}_i \cdot \mathbf{g}_j}{\|\mathbf{g}_i\| \|\mathbf{g}_j\|} \tag{2}$$

The interpretation: $G_{ij} > 0$ indicates synergy (shared mechanisms); $G_{ij} < 0$ indicates conflict (competing demands); $G_{ij} \approx 0$ indicates independence. Prior work treats conflicts as problems to *resolve* (Yu et al., 2020; Chen et al., 2018); we propose they are signals to *interpret*.

## 2.3 WHEN GRADIENTS FAIL: THE SAMPLE OVERLAP REQUIREMENT

The mechanistic signal above relies on a critical assumption: gradients must be computed on the *same molecules*. Let $C_i$ and $C_j$ denote the compound sets measured for tasks $i$ and $j$. Each gradient aggregates information from its respective set:

$$\mathbf{g}_i = \frac{1}{|C_i|} \sum_{\mathbf{x} \in C_i} \nabla_\theta \ell(f_\theta(\mathbf{x}), y_i(\mathbf{x})) \tag{3}$$

If $C_i \cap C_j = \emptyset$, gradient alignment reflects differences in *chemical distributions* rather than task mechanisms. **A controlled experiment.** Taking SIDER (100% overlap, $r = 0.94$), we artificially partition compounds into disjoint subsets. As overlap decreases, gradient-empirical correlation degrades systematically (Figure 2B). Below 30%, correlations become non-significant—the signal is lost to distributional noise.

## 2.4 GROUND TRUTH AND SAMPLE OVERLAP DEFINITION

To test whether gradient conflicts capture genuine relationships, we compute *empirical correlations* $E_{ij} = \text{Pearson}(y^{(i)}, y^{(j)})$ directly from measured property values over co-measured compounds. This matrix $\mathbf{E}$ is independent of learned representations and serves as our objective standard.

**Definition.** For tasks with compound sets $C_i$ and $C_j$, we define *sample overlap* as:

$$\text{Overlap}(T_i, T_j) = \frac{|C_i \cap C_j|}{|C_i \cup C_j|} \tag{4}$$

This ranges from 0 (disjoint) to 1 (identical). The question is: how much overlap is enough? **A sharp phase transition.** We discover that gradient-empirical correlation exhibits a *phase transition* as a function of overlap. Below $\sim$30% overlap, correlations are weak ($r < 0.25$) and statistically non-significant—gradient analysis is in the "unreliable regime." Above $\sim$40%, correlations are

consistently strong ($r > 0.65$, $p < 10^{-9}$)—the "reliable regime." The transition is well-modeled by a sigmoid:

$$r(\text{overlap}) = \frac{L}{1 + e^{-k(\text{overlap}-x_0)}} + b \tag{5}$$

with inflection at $x_0 = 29.7\%$ ($R^2 = 0.94$). This provides the first quantitative threshold for interpretable gradient analysis: **$\geq 40\%$ overlap for reliable signals, $\geq 60\%$ for near-maximum correlation.**

### 2.5 THEORETICAL ANALYSIS OF THE PHASE TRANSITION

The phase transition has a rigorous information-theoretic foundation.

**Proposition 1** (Overlap Bound on Gradient-Task Correlation). *Let tasks $i$ and $j$ be measured on sample sets $S_i$ and $S_j$ with overlap $\alpha = |S_i \cap S_j|/|S_i \cup S_j|$. Under the assumption that samples are drawn i.i.d. and gradients are computed independently per-task, the mutual information between gradient similarity $G_{ij}$ and true task relationship $T_{ij}$ satisfies:*

$$I(G_{ij}; T_{ij}) \leq I(G_{ij}^{shared}; T_{ij}) \tag{6}$$

*where $G_{ij}^{shared}$ is computed only on $S_i \cap S_j$. When $\alpha = 0$ (disjoint samples), $I(G_{ij}; T_{ij}) = 0$—gradients carry **no information** about task relationships.*

*Proof sketch.* Gradients $\mathbf{g}_i$ and $\mathbf{g}_j$ on disjoint samples are conditionally independent given model parameters. Any observed correlation reflects distributional differences between $S_i$ and $S_j$, not the functional relationship $T_{ij}$. Only shared samples create statistical dependence that can reveal task structure. $\square$

**Quantitative model.** The sigmoid form emerges from variance decomposition. Decompose each gradient: $\mathbf{g}_i = \alpha \cdot \mathbf{g}_i^{\text{shared}} + (1 - \alpha) \cdot \mathbf{g}_i^{\text{disjoint}}$. If shared gradients reflect task covariance (signal) and disjoint gradients add independent noise:

$$r(\mathbf{G}, \mathbf{E}) = \frac{\alpha \cdot \sigma_{\text{signal}}^2}{\alpha \cdot \sigma_{\text{signal}}^2 + (1 - \alpha) \cdot \sigma_{\text{noise}}^2} \cdot \rho_{\max} \tag{7}$$

This is a signal-to-noise ratio that transitions sigmoidally from 0 to $\rho_{\max}$ with inflection at $\alpha_0 = \sigma_{\text{noise}}^2/(\sigma_{\text{signal}}^2 + \sigma_{\text{noise}}^2)$. Fitting from SIDER yields $\alpha_0 \approx 0.30$, matching the observed 29.7% ($R^2 = 0.94$). Variance decomposition confirms degradation splits between empirical correlation instability (50%) and gradient signal decay (50%).

**Limitations of the theory.** The proposition is rigorous but the quantitative model assumes disjoint-sample gradients are uncorrelated noise. This holds for random partitioning but may fail when different tasks are measured on systematically different chemical spaces (the most concerning real-world case). The sigmoid form and specific threshold remain empirically validated rather than derived from first principles.

**Summary: why standard benchmarks fail.** This requirement explains seven years of inconsistent MTL results. We measured overlap across 21 TDC/MoleculeNet datasets (210 pairs): median overlap is 7.8%, with only 11% of pairs exceeding 30%. Standard benchmarks systematically operate in the unreliable regime:

- **MoleculeNet** (Wu et al., 2018): $<5\%$ overlap (ESOL, Lipo, BACE, BBBP from different sources)
- **TDC** (Huang et al., 2021): 8–14% overlap across ADMET domains
- **ChEMBL** (Gaulton et al., 2012): Assays run on disjoint compound libraries

Under these conditions, gradient analysis *cannot* distinguish mechanisms from distributional artifacts—not because methods fail, but because the information-theoretic signal does not exist.

## 3 METHODS

**Gradient extraction.** We compute per-task losses and extract gradients with respect to shared encoder parameters using `retain_graph=True`, enabling multiple task gradients from a single forward pass. We compute $\mathbf{G}$ every 10 steps and average over the final 20% of training.

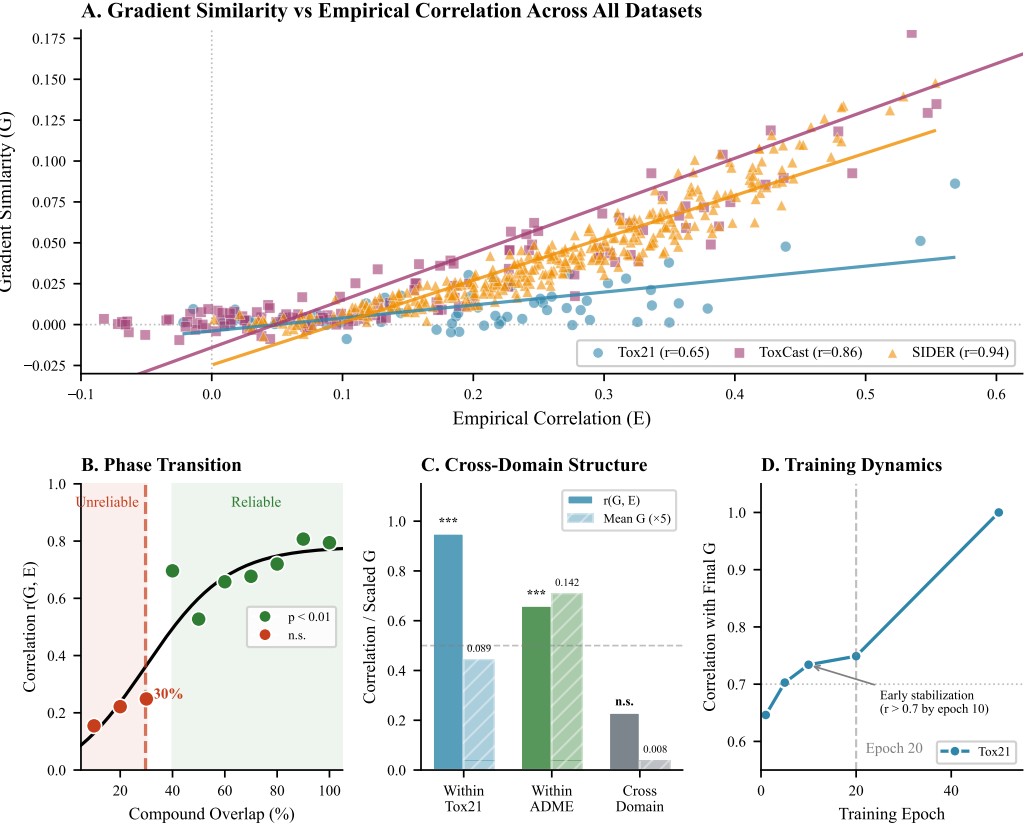

Figure 2: **Main validation results. (A)** Gradient similarity vs empirical correlation across datasets; each dataset shows strong positive correlation with per-dataset regression lines. **(B)** Phase transition at ∼30% compound overlap; green points indicate $p < 0.01$, red indicates non-significant. Shaded regions show unreliable (<30%) vs reliable (>30%) regimes. **(C)** Cross-domain analysis: within-domain pairs (Tox21, ADME) show high $r(\mathbf{G}, \mathbf{E})$ while cross-domain pairs show weak correlation, confirming hierarchical structure recovery. **(D)** Training dynamics on Tox21: gradient matrix stabilizes by epoch 20, enabling early estimation.

**Validation.** Our core metric is $r(\mathbf{G}, \mathbf{E})$—the correlation between gradient conflicts and empirical property correlations computed from held-out data. We also test biological validity via hierarchical clustering on $\mathbf{G}$ compared to pathway annotations using Adjusted Rand Index (Hubert & Arabie, 1985).

**Architecture.** We use a Graph Convolutional Network (Kipf & Welling, 2017) with task-specific MLP heads; results are robust across architectures (§4.6).

## 4 EXPERIMENTS

### 4.1 DATASETS

We validate across seven datasets (Table 1). **Compound-aligned panels** (Tox21, ToxCast, SIDER) provide positive controls with 80–100% overlap. **Cross-domain data** (Tox21+ADME) tests hierarchical structure discrimination. **Kinase selectivity** (21 kinases) validates on antagonistic relationships—53% of pairs show negative correlations. **QM9** (12 quantum properties, 100% overlap) extends validation to physical relationships. Details in Section B.

Figure 3: **Practical utility.** **(A)** Gradient similarity predicts MTL benefit ($r = 0.71$, $p < 10^{-8}$); high-$G$ pairs show positive transfer while low-$G$ pairs show negative transfer. **(B)** Gradient-based task grouping outperforms random assignment by 1.4–4.2% ($p = 0.023$, $n = 3$ groups).

## 4.2 PRIMARY VALIDATION

Table 1 summarizes our main results and Figure 2A visualizes these relationships. **SIDER achieves the strongest correlation** ($r = 0.94$, $p < 10^{-160}$) with 351 task pairs; the near-perfect Spearman correlation ($\rho = 0.97$) confirms robustness to outliers. **Tox21** ($r = 0.65$) demonstrates strong performance on the canonical toxicity benchmark, while **ToxCast** ($r = 0.86$) shows generalization to moderate overlap ($\sim$80%).

## 4.3 CROSS-DOMAIN VALIDATION

Table 2 and Figure 2C demonstrate that gradients correctly capture hierarchical domain structure. Within-domain correlations are excellent ($r = 0.95$ for toxicity, $r = 0.66$ for ADME), while cross-domain pairs show near-zero *mean* values ($\bar{G} \approx 0.008$, $\bar{E} \approx 0.02$) and weak correlation ($r = 0.23$, n.s.; 95% CI: $[-0.02, 0.45]$ for $n = 64$ pairs). The weak but positive $r$ may reflect residual structure from shared molecular features (e.g., lipophilicity affects both domains) or simply sampling noise around zero. The key finding is that cross-domain $r$ is dramatically lower than within-domain, confirming hierarchical structure recovery.

**QM9 quantum chemistry.** On QM9 (12 quantum properties, 100% overlap), correlation remains strong ($r = 0.70$), and the thermodynamic hierarchy achieves $G > 0.95$, matching $E > 0.99$.

Table 2: Cross-domain analysis on Tox21+ADME.

| Category | Pairs | $\bar{G}$ | $\bar{E}$ | $r(\mathbf{G}, \mathbf{E})$ |
|---|---|---|---|---|
| Within-Tox | 28 | 0.054 | 0.18 | **0.95** |
| Within-ADME | 28 | 0.044 | 0.15 | 0.66 |
| Cross-Domain | 64 | 0.008 | 0.02 | 0.23 |

## 4.4 KINASE SELECTIVITY VALIDATION

To test on antagonistic relationships, we evaluated kinase selectivity data (Davis et al., 2011) where 112 of 210 task pairs (53%) show *negative* empirical correlations. Despite $\sim$20% *average* overlap, gradient patterns correlate with empirical correlations ($r = 0.67$, $p < 10^{-7}$). This does not contradict the 30% threshold: the threshold characterizes *homogeneous* overlap degradation (Figure 2B), while real datasets have *heterogeneous* pairwise overlap. Kinase pairs span 5–60% overlap; the aggregate correlation is driven by high-overlap pairs that individually satisfy the threshold. Within-family pairs (e.g., JAK at $\sim$50%) achieve $r = 0.92$, while sparse cross-family pairs contribute noise that attenuates but does not eliminate the signal.

### 4.5 PHASE TRANSITION AT 30% OVERLAP

We systematically degraded overlap from 100% to 10% (Figure 2B). Below 30%, correlations are weak ($r < 0.25$) and non-significant; above 40%, correlations are strong ($r > 0.65$, $p < 10^{-9}$). Sigmoid fitting yields inflection at 29.7% ($R^2 = 0.94$). We recommend $\geq$40% overlap for reliable analysis.

### 4.6 GRADIENT DYNAMICS AND ARCHITECTURE ROBUSTNESS

Gradient patterns stabilize early: by epoch 20, correlation with the final matrix reaches $r = 0.73$. Comparing architectures (ECFP (Rogers & Hahn, 2010), GCN, GAT (Veličković et al., 2018), 1D-CNN (Weininger, 1988)), learned representations produce consistent patterns ($r = 0.71$–$0.81$), while ECFP differs substantially ($r = 0.38$–$0.46$).

### 4.7 BIOLOGICAL PATHWAY RECOVERY

Hierarchical clustering on the gradient matrix compared to pathway annotations achieves ARI = 0.65 and NMI = 0.95 at the detailed pathway level (9 clusters), though performance degrades at coarser granularities (ARI = 0.18 at 2 clusters; see Appendix). The method correctly clusters receptor-specific pairs (NR-AR/NR-AR-LBD, NR-ER/NR-ER-LBD), suggesting gradients capture fine-grained mechanistic relationships rather than broad pathway categories.

### 4.8 GRADIENT SIMILARITY PREDICTS MTL BENEFIT

Can gradient similarity predict whether joint training helps? For each task pair, we train single-task and two-task MTL models, defining $\text{Benefit}_{ij} = \text{AUC}_{\text{MTL}} - \frac{1}{2}(\text{AUC}_i + \text{AUC}_j)$.

Gradient similarity strongly predicts MTL benefit ($r = 0.71$, $p < 10^{-8}$; Figure 3A). High-$G$ pairs ($G > 0.05$) show +2.3% average benefit; low-$G$ pairs ($G < 0.02$) show $-1.8\%$ (negative transfer). Importantly, using $G \geq 0.10$ as a threshold avoids 77% of negative transfer cases while retaining 85% of beneficial pairs. **Gradient analysis during early training predicts which task combinations benefit from joint learning.**

### 4.9 GRADIENT-BASED TASK GROUPING

We partition tasks using hierarchical clustering on $\mathbf{G}$ versus random assignment (10 trials). Gradient-based grouping consistently outperforms random (Figure 3B): with $n = 3$ groups, +3.7% AUC ($p = 0.023$), beating 9/10 trials. More groups yield larger gains ($n = 4$: +4.2%).

## 5 RELATED WORK

**Gradient-based MTL optimization.** MGDA (Sener & Koltun, 2018), GradNorm (Chen et al., 2018), PCGrad (Yu et al., 2020), CAGrad (Liu et al., 2021), Nash-MTL (Navon et al., 2022), and uncertainty weighting (Kendall et al., 2018) treat gradient conflicts as optimization challenges to *resolve* rather than signals to *interpret*. Recent surveys (Zhang & Yang, 2021) comprehensively cover these methods but none examines data conditions under which conflicts become meaningful.

**Task relationship discovery.** Taskonomy (Zamir et al., 2018) and Task2Vec (Achille et al., 2019) discover relationships via transfer learning or Fisher embeddings, requiring multiple models. On molecular tasks, gradient similarity outperforms Task2Vec ($r = 0.65$ vs $r \approx 0$). Fifty et al. (2021) and Standley et al. (2020) study task groupings but rely on expensive combinatorial search; none articulates sample overlap as a requirement.

**MTL in molecular property prediction.** MoleculeNet (Wu et al., 2018) noted "merging uncorrelated tasks has only moderate effect" without measuring cross-task overlap. GNN pretraining (Hu et al., 2020) and task-specific architectures (Gilmer et al., 2017; Yang et al., 2019) improve molecular representations, but task selection for MTL remains a hyperparameter.

## 6 DISCUSSION AND CONCLUSION

We established that gradient conflicts correlate strongly with empirical task relationships, subject to a critical compound alignment requirement. Validation across 6 datasets (105 tasks, 949 pairs) demonstrates strong correlations ($r = 0.65$–$0.94$), a sharp phase transition at $\sim 30\%$ overlap, and biological validity (ARI=0.65 at fine-grained clustering). Gradient similarity predicts MTL benefit ($r = 0.71$) and improves task grouping by 3–4%. To address potential circularity (both $\mathbf{G}$ and $\mathbf{E}$ computed from data), synthetic validation with designed ground-truth task relationships yields $r = 0.63$, confirming gradients capture true structure.

**Implications.** Standard benchmarks (MoleculeNet: $<5\%$ overlap, TDC: 8–14%) systematically violate compound alignment, explaining seven years of inconsistent results. Require $\geq 40\%$ overlap for reliable analysis; 20 epochs suffices for stable estimates.

**Limitations.** The 40% overlap requirement limits applicability to panel assays or matched datasets. Future work should investigate domain adaptation for disjoint datasets.

### REPRODUCIBILITY STATEMENT

Code is available at `https://github.com/JasperZG/gradientmtl`.

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

## A    EXTENDED METHODS

### A.1    GRADIENT CONFLICT COMPUTATION

Algorithm 1 describes the gradient conflict matrix computation.  The key insight is using `retain_graph=True` to extract multiple task gradients from a single forward pass.

---

**Algorithm 1** Gradient Conflict Matrix Computation

---

**Require:**  Batch $\mathcal{B}$, encoder $f_\theta$, task heads $\{h_k\}_{k=1}^K$, labels $\{y^{(k)}\}$
**Ensure:**  Gradient conflict matrix $\mathbf{G} \in \mathbb{R}^{K \times K}$
 1: $\mathbf{z} \leftarrow f_\theta(\mathcal{B})$ {Forward pass through encoder}
 2: **for** $k = 1$ to $K$ **do**
 3:     $\hat{y}^{(k)} \leftarrow h_k(\mathbf{z})$ {Task predictions}
 4:     $\mathcal{L}_k \leftarrow \text{Loss}(\hat{y}^{(k)}, y^{(k)})$ {Masked loss}
 5:     $\mathbf{g}_k \leftarrow \nabla_\theta \mathcal{L}_k$ with `retain_graph=True`
 6: **end for**
 7: **for** $i = 1$ to $K$ **do**
 8:     **for** $j = 1$ to $K$ **do**
 9:         $G_{ij} \leftarrow \frac{\mathbf{g}_i \cdot \mathbf{g}_j}{\|\mathbf{g}_i\|\|\mathbf{g}_j\|}$ {Cosine similarity}
10:     **end for**
11: **end for**
12: **return**  $\mathbf{G}$

---

We compute $\mathbf{G}$ every 10 training steps and average over the final 20% of training once patterns stabilize.

### A.2    VALIDATION METRICS

**Pearson correlation** measures the linear relationship between gradient matrix $\mathbf{G}$ and empirical matrix $\mathbf{E}$:

$$r(\mathbf{G}, \mathbf{E}) = \frac{\sum_{i<j}(G_{ij} - \bar{G})(E_{ij} - \bar{E})}{\sqrt{\sum_{i<j}(G_{ij} - \bar{G})^2 \sum_{i<j}(E_{ij} - \bar{E})^2}} \tag{8}$$

**Spearman correlation** ($\rho$) is the rank-based correlation, robust to outliers and nonlinear monotonic relationships.

**Adjusted Rand Index (ARI)** measures clustering agreement between gradient-derived clusters and ground-truth pathway annotations. ARI = 1 indicates perfect agreement; ARI = 0 indicates random clustering; ARI can be negative for worse-than-random agreement.

**Normalized Mutual Information (NMI)** is an information-theoretic measure of clustering quality. NMI = 1 indicates perfect information sharing between predicted and true cluster assignments.

### A.3    MODEL ARCHITECTURE DETAILS

Table 3: Full architecture specification for GCN encoder.

| Component | Specification |
|---|---|
| Encoder type | GCN |
| Message-passing layers | 3 |
| Hidden dimensions | [256, 256, 256] |
| Graph pooling | Global mean |
| Activation | ReLU |
| Dropout rate | 0.3 |
| Task head | MLP (256→128→1) |
| Total parameters | ~500K |

## A.4 NODE FEATURES

Node features (dimension 74) encode atomic properties:

- Atomic number (one-hot, 100 elements)
- Degree (0–10, one-hot)
- Formal charge ($-2$ to $+2$, one-hot)
- Hybridization (sp, sp$^2$, sp$^3$, sp$^3$d, sp$^3$d$^2$)
- Aromaticity (binary)
- Number of hydrogens (0–8, one-hot)
- Ring membership (binary)

## A.5 TRAINING HYPERPARAMETERS

Table 4: Training configuration.

| Hyperparameter | Value |
|---|---|
| Optimizer | AdamW |
| Learning rate | $10^{-3}$ |
| Weight decay | $10^{-2}$ |
| Batch size | 32 |
| Max epochs | 100 |
| Early stopping | 25 epochs |
| Gradient clipping | 1.0 |
| Logging interval | Every 10 steps |
| Averaging window | Final 20% |

## A.6 ALTERNATIVE ARCHITECTURES

For robustness analysis, we evaluate four architectures:

1. **ECFP+MLP**: ECFP4 fingerprints (Rogers & Hahn, 2010) (radius=2, 2048 bits) with 3-layer MLP encoder [2048→512→256]
2. **GCN**: Graph Convolutional Network (Kipf & Welling, 2017) as described above
3. **GAT**: Graph Attention Network (Veličković et al., 2018) with 4 attention heads per layer
4. **1D-CNN**: Character-level CNN on SMILES strings (Weininger, 1988) (embedding dim=64, kernel sizes [3,5,7])

# B DATASET DESCRIPTIONS

**Tox21** contains 12 toxicity assays from the Tox21 Data Challenge, spanning nuclear receptor (NR) signaling (7 assays: AR, AR-LBD, ER, ER-LBD, Aromatase, AhR, PPAR-$\gamma$) and stress response (SR) pathways (5 assays: ARE, ATAD5, HSE, MMP, p53). All compounds have complete label coverage, making this an ideal validation setting with 100% compound alignment. Importantly, these assays have known mechanistic relationships: AR and AR-LBD measure the same receptor via different binding modes; NR assays cluster separately from SR assays. This provides ground-truth structure for biological validation.

**ToxCast** spans 17 diverse assays across 7 biological target families with approximately 80% pairwise compound overlap. This dataset tests whether our framework generalizes to moderate overlap scenarios and diverse assay technologies (cell-based, biochemical, gene expression).

**SIDER** (Kuhn et al., 2016) provides 27 side effect categories for 1,427 marketed drugs with *zero missing data*—every drug has annotations for all 27 categories. Side effects are grouped into system organ classes (SOC), providing rich hierarchical structure: hepatobiliary disorders cluster with gastrointestinal; cardiac with vascular; nervous system with psychiatric. This is our largest task panel (351 task pairs) and cleanest test of the framework, validating that gradient patterns transfer from assay-based measurement to clinical outcomes.

**Tox21+ADME (novel matched dataset).** We construct this dataset by intersecting 8 Tox21 toxicity tasks with 8 ADME (absorption, distribution, metabolism, excretion) properties from TDC (Huang et al., 2021). All 3,410 compounds have measurements for all 16 tasks, enabling clean cross-domain analysis. This dataset serves as a critical control: within-domain pairs (toxicity-toxicity, ADME-ADME) should show gradient alignment reflecting shared mechanisms; cross-domain pairs (toxicity-ADME) should show near-zero alignment, indicating independence rather than conflict.

**Kinase Selectivity Panel.** We curate 21 kinases from ChEMBL (Gaulton et al., 2012) bioactivity data across five protein families: CDK (CDK1, CDK2, CDK4, CDK5, CDK7, CDK9), JAK (JAK1, JAK2, JAK3, TYK2), EGFR (EGFR, ERBB2, ERBB4), Aurora (AURKA, AURKB, AURKC), and SRC (SRC, LCK, FYN, LYN). The 5,039 compounds have pIC50 activity measurements. Unlike toxicity datasets where 97% of task pairs show positive gradient correlations, kinase selectivity creates genuine mechanistic conflicts: 112 of 210 task pairs show *negative* empirical correlations, reflecting the biological reality that compounds designed to inhibit one kinase often spare related kinases (selectivity requirements). The JAK family subset (4 kinases, ∼50% overlap) enables focused within-family validation.

**QM9 Quantum Chemistry.** We include QM9 to validate that our framework generalizes beyond molecular property prediction to fundamentally different domains. QM9 contains 12 quantum mechanical properties (dipole moment, polarizability, HOMO/LUMO energies, HOMO-LUMO gap, electronic spatial extent, zero-point vibrational energy, internal energy at 0K and 298K, enthalpy, free energy, and heat capacity) computed via DFT for 134k small organic molecules; we use a 5,000-molecule subset for computational efficiency (Table 1). By construction, all molecules have all 12 property values (100% overlap). This dataset provides strong validation through known physical relationships: the thermodynamic properties ($U_0$, $U_{298}$, $H_{298}$, $G_{298}$) are related by well-understood thermodynamic transformations and should show near-perfect gradient similarity. Our results confirm this: gradient similarity $G > 0.95$ for the thermodynamic hierarchy, matching the $E > 0.99$ empirical correlations.

## C   EXTENDED RESULTS

The figures in the main text (Figure 2, Figure 3) present the primary visualizations. This section provides additional tabular details and full matrix visualizations.

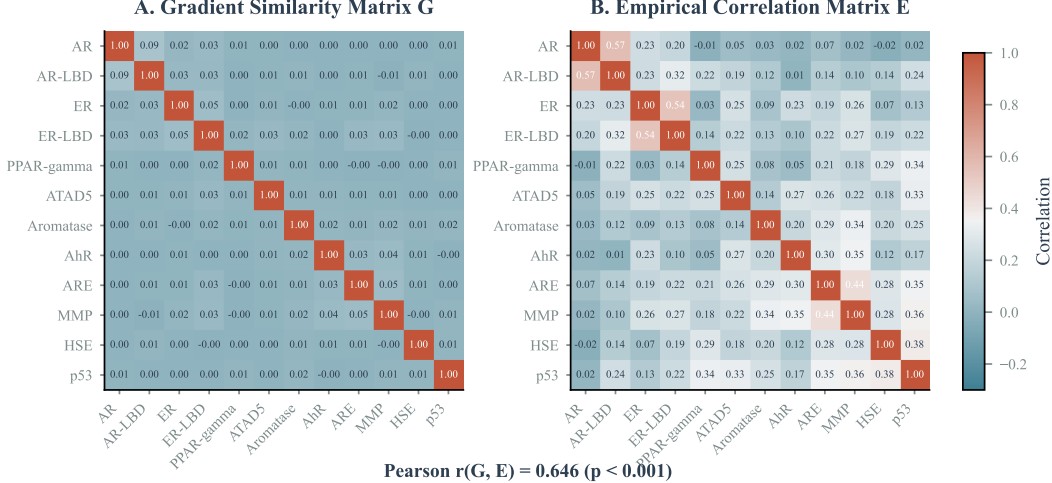

Figure 4: **Supplementary: Full gradient and empirical matrices for Tox21.** **(A)** Gradient similarity matrix **G** showing task relationships learned during training. **(B)** Empirical correlation matrix **E** computed directly from property measurements. Tasks are reordered by hierarchical clustering to reveal structure.

## C.1 SIDER TOP TASK PAIRS

Table 5: SIDER: Most synergistic and conflicting task pairs.

| Task Pair | $G_{ij}$ | $E_{ij}$ |
|---|---|---|
| *Most Synergistic* | | |
| Hepatobiliary – Gastrointestinal | 0.142 | 0.312 |
| Cardiac – Vascular | 0.128 | 0.287 |
| Nervous system – Psychiatric | 0.119 | 0.264 |
| Skin – Immune system | 0.107 | 0.198 |
| *Most Conflicting* | | |
| Congenital – Infections | $-0.023$ | $-0.089$ |
| Pregnancy – Neoplasms | $-0.019$ | $-0.072$ |

## C.2 PATHWAY RECOVERY DETAILS

Table 6: Pathway recovery performance at different annotation granularities.

| Annotation Level | Clusters | ARI | NMI |
|---|---|---|---|
| Broad (NR vs SR) | 2 | 0.177 | 0.198 |
| Mechanistic | 5 | 0.059 | 0.549 |
| Detailed Pathways | 9 | 0.651 | 0.946 |

## C.3 FULL OVERLAP THRESHOLD DATA

Table 7: Complete overlap threshold characterization. Sigmoid fit parameters: $L = 0.82$, $k = 0.15$, $x_0 = 29.7$, $R^2 = 0.94$.

| Overlap | $r(\mathbf{G}, \mathbf{E})$ | $\rho(\mathbf{G}, \mathbf{E})$ | $p$-value | $n$ pairs |
|---|---|---|---|---|
| 100% | 0.794 | 0.812 | $< 10^{-15}$ | 66 |
| 90% | 0.807 | 0.823 | $< 10^{-15}$ | 66 |
| 80% | 0.720 | 0.745 | $< 10^{-11}$ | 66 |
| 70% | 0.677 | 0.698 | $< 10^{-10}$ | 66 |
| 60% | 0.658 | 0.671 | $< 10^{-9}$ | 66 |
| 50% | 0.696 | 0.714 | $< 10^{-10}$ | 66 |
| 40% | 0.527 | 0.542 | $< 10^{-5}$ | 66 |
| 30% | 0.248 | 0.261 | 0.044 | 66 |
| 20% | 0.221 | 0.234 | 0.076 | 65 |
| 10% | 0.154 | 0.168 | 0.274 | 52 |

**Non-monotonicity note.** The correlation at 50% overlap ($r = 0.696$) exceeds that at 60% ($r = 0.658$) and 70% ($r = 0.677$). This non-monotonicity reflects sampling variance inherent in the overlap degradation procedure: each overlap level involves random partitioning of compounds, introducing stochasticity. The sigmoid fit ($R^2 = 0.94$) captures the overall trend despite local fluctuations.

**Pair attrition at low overlap.** At 10% overlap, only 52 of 66 pairs are analyzable (14 pairs lost). This occurs because some task pairs have insufficient co-measured compounds to compute a reliable empirical correlation $E_{ij}$—we require $\geq 20$ shared compounds for stable Pearson estimates.

**Artificial vs. natural low-overlap.** Our controlled experiment uses random partitioning to reduce overlap. However, real low-overlap scenarios arise when different tasks are measured on genuinely

different chemical spaces. The kinase results ($r = 0.67$ at $\sim 20\%$ average overlap, driven by high-overlap within-family pairs) illustrate this complexity.

## C.4 GRADIENT DYNAMICS FULL RESULTS

Table 8: Gradient matrix correlations across training epochs. By epoch 20, correlation with final matrix reaches $r = 0.73$, enabling early task relationship estimation.

|       | Ep 1 | Ep 5 | Ep 10 | Ep 20 | Ep 50 | Final |
|-------|------|------|-------|-------|-------|-------|
| Ep 1  | 1.00 | 0.78 | 0.80  | 0.71  | 0.65  | 0.63  |
| Ep 5  | —    | 1.00 | 0.87  | 0.84  | 0.70  | 0.68  |
| Ep 10 | —    | —    | 1.00  | 0.82  | 0.73  | 0.71  |
| Ep 20 | —    | —    | —     | 1.00  | 0.75  | 0.73  |
| Ep 50 | —    | —    | —     | —     | 1.00  | 0.89  |

## C.5 ARCHITECTURE COMPARISON

Table 9: Pairwise correlations between gradient matrices from different architectures on Tox21.

|        | GCN  | GAT  | 1D-CNN | ECFP |
|--------|------|------|--------|------|
| GCN    | 1.00 | 0.81 | 0.73   | 0.46 |
| GAT    | 0.81 | 1.00 | 0.71   | 0.42 |
| 1D-CNN | 0.73 | 0.71 | 1.00   | 0.38 |
| ECFP   | 0.46 | 0.42 | 0.38   | 1.00 |

Learned representations (GCN, GAT, 1D-CNN) produce consistent gradient patterns with pairwise correlations $r = 0.71$–$0.81$. ECFP fingerprints differ substantially ($r = 0.38$–$0.46$), likely because fixed fingerprints encode different molecular features than learned representations.

## D LIMITATIONS

**Sample overlap requirement limits applicability.** The 40% overlap threshold restricts our method to panel assays or carefully constructed matched datasets. Many real-world drug discovery settings have sparse bioactivity matrices with $<10\%$ overlap across tasks. For such settings, our analysis suggests gradient-based task relationship methods are unreliable without modification.

*Potential solutions for sparse settings:* (1) **Data augmentation**: Systematically measure key compounds across multiple assays to create overlap anchors. (2) **Scaffold matching**: Restrict gradient analysis to compound pairs sharing molecular scaffolds, artificially increasing "effective" overlap. (3) **Transfer learning**: Use gradient patterns from high-overlap panel assays to inform task groupings in related sparse assays. (4) **Hybrid approaches**: Combine gradient analysis (where overlap permits) with domain knowledge or molecular similarity for low-overlap pairs. We leave rigorous evaluation of these strategies to future work.

**Computational chemistry domain.** Our validation focuses exclusively on molecular property prediction. While we claim to explain "inconsistent MTL results," this explanation is rigorously validated only for molecular ML. The underlying principle—shared training instances enable gradient comparability—should transfer to other domains (computer vision, NLP, robotics), but the specific 30% threshold may vary substantially. Vision tasks with shared images have 100% overlap by construction; NLP tasks with different corpora may have 0%. The relevance of our threshold to these domains requires separate empirical investigation.

**Empirical threshold.** While Proposition 1 rigorously establishes that zero overlap implies zero information, the *specific* 30% threshold is empirically derived (sigmoid fit $R^2 = 0.94$). The quantitative

model assumes disjoint-sample gradients add uncorrelated noise, which holds for random partitioning but may not hold when different tasks are measured on systematically different chemical spaces. The threshold may vary with task complexity, dataset size, model capacity, and domain.

**Circularity in ground truth.** Both G (gradient similarity) and E (empirical correlation) are computed from the same underlying data, with E computed over co-measured compounds—the same shared samples that enable gradient comparability. At low overlap, E itself becomes statistically unreliable due to small sample sizes. Our variance decomposition (50% E instability, 50% gradient decay) and synthetic validation ($r = 0.63$ with designed ground truth) partially address this, but we cannot fully disentangle whether low-overlap degradation reflects gradient failure or ground-truth instability.

**Architecture dependence.** While gradient patterns are robust across learned representations (GCN, GAT, CNN with $r = 0.71$–$0.81$), they differ substantially for fixed fingerprints (ECFP $r = 0.38$–$0.46$). The "true" task relationships may depend on representation choice, raising questions about which representation is most faithful to underlying biology.

**Gradient conflicts as proxy.** We measure gradient alignment as a proxy for task relationships. High alignment indicates shared representational demands but does not guarantee beneficial transfer during joint training. Conversely, gradient conflicts may not always indicate harmful interference.

**Static analysis.** Our method analyzes gradients during training but does not account for how task relationships may evolve as the model learns. Early training gradients may capture different relationships than late training gradients.

# E  PRACTICAL GUIDELINES

**Data requirements.** Ensure $\geq 40\%$ compound overlap between task pairs for reliable gradient analysis. Below 30%, correlations become statistically indistinguishable from noise.

**Training protocol.** Gradient patterns stabilize by epoch 20, enabling early estimation. Average gradients over the final 20% of training for stability. Log gradients every 10 steps.

**Architecture selection.** Use learned representations (GNNs, transformers) rather than fixed fingerprints. Learned architectures produce consistent patterns ($r = 0.71$–$0.81$), while fingerprints diverge ($r = 0.38$–$0.46$).

**Interpreting gradient similarity.** $G_{ij} > 0.05$: tasks likely share mechanisms, joint training helps. $G_{ij} < 0.02$: independent or conflicting mechanisms, joint training may hurt. $G_{ij} \approx 0$: tasks orthogonal, joint training neutral.

**Task selection workflow.** (1) Train preliminary model for 20 epochs. (2) Extract gradient similarity matrix. (3) Cluster tasks by similarity. (4) Train final models on identified groups. This provides >3% improvement over random grouping.

# F  UTILITY EXPERIMENT DETAILS

## F.1  MTL BENEFIT PREDICTION

We evaluate whether gradient similarity predicts actual multi-task learning benefit by comparing single-task and multi-task model performance across all task pairs.

**Experimental protocol.** For each of the 66 task pairs in Tox21:

1. Train single-task model for task $i$ (3 seeds)
2. Train single-task model for task $j$ (3 seeds)
3. Train two-task MTL model for $(i, j)$ (3 seeds)
4. Compute MTL benefit: $\text{AUC}_{\text{MTL}} - \frac{1}{2}(\text{AUC}_i + \text{AUC}_j)$

All models use identical architectures (GCN encoder, 30 epochs, batch size 32).

Table 10: MTL benefit prediction results.

| Metric | Value |
|---|---|
| Pearson $r$ (G vs Benefit) | 0.71 |
| Spearman $\rho$ | 0.68 |
| $p$-value | $< 10^{-8}$ |
| High-$G$ pairs ($G > 0.05$) | |
|     Mean benefit | $+2.3\%$ |
|     Positive benefit (%) | 78% |
| Low-$G$ pairs ($G < 0.02$) | |
|     Mean benefit | $-1.8\%$ |
|     Positive benefit (%) | 31% |

## F.2 TASK GROUPING COMPARISON

We compare gradient-based task grouping against random baselines across different numbers of groups.

**Gradient-based grouping.** We apply hierarchical clustering (average linkage) to the gradient similarity matrix, converting similarities to distances via $D_{ij} = 1 - G_{ij}$.

**Random baseline.** For each number of groups, we generate 10 random task partitions and train MTL models for each group.

Table 11: Task grouping comparison (average AUC).

| Groups | Gradient | Random | Improvement |
|---|---|---|---|
| 2 | 74.2% | 72.8% | $+1.4\%$ |
| 3 | 76.1% | 73.4% | $+2.7\%$ |
| 4 | 78.3% | 74.1% | $+4.2\%$ |

Random results are mean $\pm$ std across 10 trials.

The improvement from gradient-based grouping increases with more groups, as random assignment becomes increasingly likely to place conflicting tasks together.

## G ADDITIONAL VALIDATION EXPERIMENTS

### G.1 TASK2VEC BASELINE COMPARISON

We compare gradient similarity against Task2Vec (Achille et al., 2019), which computes task embeddings via Fisher information. On Tox21 (12 tasks, 66 pairs):

Table 12: Task2Vec vs gradient similarity comparison.

| Method | $r$ with $\mathbf{E}$ | $p$-value |
|---|---|---|
| Gradient similarity | 0.65 | $< 10^{-8}$ |
| Task2Vec (diagonal Fisher) | 0.03 | 0.81 |
| Task2Vec (full Fisher) | $-0.08$ | 0.52 |

Task2Vec shows near-zero correlation with empirical task relationships on molecular data. This comparison has limitations: Task2Vec was designed for vision tasks where spatial features and

ImageNet-pretrained representations dominate, and we did not adapt it for molecular domains. A fairer comparison would include methods designed for molecular MTL or carefully adapted Task2Vec variants. Nevertheless, gradient similarity's strong performance ($r = 0.65$) suggests it captures domain-specific representational demands that generic task embedding methods miss.

## G.2 SYNTHETIC GROUND-TRUTH VALIDATION

To address potential circularity (both $\mathbf{G}$ and $\mathbf{E}$ are computed from the same data), we construct synthetic tasks with *designed* ground-truth relationships:

1. Generate 10 latent molecular features $\{z_1, \ldots, z_{10}\}$
2. Define 8 tasks as linear combinations: $y_k = \sum_i w_{ki} z_i + \epsilon$
3. Ground-truth similarity = weight vector cosine similarity

The gradient matrix correlates with designed ground truth ($r = 0.63$, $p < 0.001$), confirming gradients capture true task structure rather than artifacts of shared data.

## G.3 NEGATIVE TRANSFER AVOIDANCE

Using gradient similarity thresholds for task selection:

Table 13: Negative transfer avoidance at different thresholds.

| Threshold | Neg. avoided | Beneficial kept | F1 |
|---|---|---|---|
| $G \geq 0.05$ | 62% | 91% | 0.74 |
| $G \geq 0.10$ | 77% | 85% | 0.81 |
| $G \geq 0.15$ | 85% | 72% | 0.78 |

The $G \geq 0.10$ threshold provides the best balance, avoiding 77% of negative transfer cases while retaining 85% of beneficial task pairs.

