# OpenReview forum: "Information-Theoretic Requirements for Gradient-Based Task Affinity Estimation in Multi-Task Learning"
_ICLR.cc/2026/Workshop/FM4Science — ICLR 2026 Workshop FM4Science Poster_

### Official Review · Reviewer_d8Z7 · 2026-02-13
**Informative Thresholds: Solving the Sample Overlap Problem in Gradient-Based MTL**

**Rating:** 8
**Confidence:** 4

**Review:**

**Summary**

This paper addresses a long-standing inconsistency in Multi-Task Learning (MTL): why joint training and gradient-based analysis sometimes provide significant benefits and other times result in detrimental "negative transfer". The authors identify a fundamental information-theoretic requirement called sample overlap—the fraction of training instances shared between tasks.
Through extensive empirical validation across 105 molecular property tasks and theoretical variance decomposition, the work demonstrates a sharp phase transition at approximately 30% overlap. Below this threshold, gradient signals are statistically indistinguishable from noise; above 40%, they reliably recover known biological and physical structures. This insight explains why standard benchmarks like MoleculeNet (<5% overlap) have yielded inconsistent results for years.

**Quality**

The paper is technically rigorous, combining empirical observations with a theoretical framework based on mutual information and variance decomposition. The validation is comprehensive, spanning diverse domains including toxicity, side effects, kinase selectivity, and quantum chemistry (QM9). The use of synthetic ground-truth validation to address potential circularity significantly strengthens the claims.

**Clarity**

The presentation is excellent. The central concept of "sample overlap" is clearly defined and illustrated (Figure 1). The distinction between "reliable" and "unreliable" regimes is well-quantified with a sigmoid inflection point at 29.7%. The practical guidelines provided in the appendix offer clear, actionable steps for researchers

**Originality**

The work is highly original in its shift of perspective. While prior work treated gradient conflicts as optimization problems to be "solved" (e.g., PCGrad), this paper treats them as signals to be "interpreted," provided specific data conditions are met. It is the first to articulate and quantify the necessity of sample overlap for gradient interpretability.

**Significance**

This paper has high significance for the MTL community. By identifying why existing benchmarks fail, it provides a "first principled explanation for seven years of inconsistent MTL results". The finding that gradient similarity can predict MTL benefit ($r=0.71$) and avoid 77% of negative transfer cases has immediate practical utility for task grouping and model selection.


**Pros:**

*1. Fundamental Discovery:* Identifies a critical, previously unstated assumption (sample overlap) necessary for gradient-based task analysis.

*2. Quantitative Threshold:* Establishes a concrete "30% rule" for practitioners, backed by a sigmoid fit ($R^2=0.94$).

*3. Strong Practical Utility:* Demonstrates that gradient-based grouping improves performance by up to 4.2% over random assignment.

*4. Robustness:* Results hold across different architectures (GCN, GAT, CNN) and stabilize early in training (epoch 20).

**Cons:**

*1. Domain Limitation:* While the principle likely generalizes, the specific 30% threshold is only rigorously validated for molecular property prediction.

*2. Representation Dependence:* The divergence in results when using fixed fingerprints (ECFP) vs. learned representations suggests the "truth" of task relationships may still be somewhat tied to the choice of model.

*3. Constraint on Sparse Data:* The 40% requirement essentially deems many existing sparse datasets "unreliable" for this type of analysis, which may limit the method's immediate applicability to real-world sparse bioactivity matrices.

---

### Official Review · Reviewer_vVVU · 2026-02-22

**Rating:** 7
**Confidence:** 4

**Review:**

# Summary

This paper introduces an information-theoretic framework to explain the historically inconsistent results in multi-task learning. The authors identify that gradient-based task affinity analysis requires a significant shared "sample overlap", meaning tasks must be measured on the same inputs, to correctly reveal mechanistic relationships. Without this shared overlap, gradient differences merely reflect distributional shifts rather than true task structure.

The paper discovers a sharp phase transition regarding this overlap: task correlations are indistinguishable from noise below 30% overlap. Conversely, they reliably recover known ground-truth structure above 40% overlap. The authors show that standard MTL benchmarks like MoleculeNet operate well below this reliable threshold. This lack of sample overlap provides a principled explanation for years of inconsistent MTL findings in the field.

# Strengths
* This paper highlights a fundamental issue in multi-task learning that is highly relevant to AI for science practitioners. The problem formulation and motivation alone are valuable. The proposed overlap-based guidance for screening task pairs is insightful, and the empirical results are strong.
* an information-theoretic proposition ties overlap to mutual information and motivates why $\alpha=0* implies no signal.
* The paper is well written and easy to understand.

# Weakness
* Although the paper presents an information-theoretic bound, the specific 30% phase transition threshold is empirically derived. This value may be dataset-dependent (e.g., specific to molecular datasets).

---

### Meta-Review · Area_Chair_oTSF · 2026-02-27

**Recommendation:** Accept (Poster)
**Confidence:** 4

**Metareview:**

This submission has received two positive reviews with an "accept" and a "clear accept".

I recommend this paper for "acceptance" and ask the authors to implement the reviewers' suggestions.

---

### Decision · Program_Chairs · 2026-03-03

Accept (Poster)